# On Adaptive Distance Estimation

**Yeshwanth Cherapanamjeri**
Electrical Engineering and Computer Science
University of California at Berkeley
Berkeley, CA 94720
yeshwanth@berkeley.edu

**Jelani Nelson**
Electrical Engineering and Computer Science
University of California at Berkeley
Berkeley, CA 94720
minilek@berkeley.edu

## Abstract

We provide a static data structure for distance estimation which supports *adaptive* queries. Concretely, given a dataset $X = \{x_i\}_{i=1}^n$ of $n$ points in $\mathbb{R}^d$ and $0 < p \le 2$, we construct a randomized data structure with low memory consumption and query time which, when later given any query point $q \in \mathbb{R}^d$, outputs a $(1 + \varepsilon)$-approximation of $\|q - x_i\|_p$ with high probability for all $i \in [n]$. The main novelty is our data structure's correctness guarantee holds even when the sequence of queries can be chosen adaptively: an adversary is allowed to choose the $j$th query point $q_j$ in a way that depends on the answers reported by the data structure for $q_1, \ldots, q_{j-1}$. Previous randomized Monte Carlo methods do not provide error guarantees in the setting of adaptively chosen queries [JL84, Ind06, TZ12, IW18]. Our memory consumption is $\tilde{O}((n + d)d/\varepsilon^2)$, slightly more than the $O(nd)$ required to store $X$ in memory explicitly, but with the benefit that our time to answer queries is only $\tilde{O}(\varepsilon^{-2}(n + d))$, much faster than the naive $\Theta(nd)$ time obtained from a linear scan in the case of $n$ and $d$ very large. Here $\tilde{O}$ hides $\log(nd/\varepsilon)$ factors. We discuss applications to nearest neighbor search and nonparametric estimation.

Our method is simple and likely to be applicable to other domains: we describe a generic approach for transforming randomized Monte Carlo data structures which do not support adaptive queries to ones that do, and show that for the problem at hand, it can be applied to standard nonadaptive solutions to $\ell_p$ norm estimation with negligible overhead in query time and a factor $d$ overhead in memory.

## 1 Introduction

In recent years, much research attention has been directed towards understanding the performance of machine learning algorithms in adaptive or adversarial environments. In diverse application domains ranging from malware and network intrusion detection [BCM+17, CBK09] to strategic classification [HMPW16] to autonomous navigation [PMG16, LCLS17, PMG+17], the vulnerability of machine learning algorithms to malicious manipulation of input data has been well documented. Motivated by such considerations, we study the problem of designing efficient data structures for distance estimation, a basic primitive in algorithms for nonparametric estimation and exploratory data analysis, in the adaptive setting where the sequence of queries made to the data structure may be adversarially chosen. Concretely, the distance estimation problem is defined as follows:

**Problem 1.1** (Approximate Distance Estimation (ADE)). For a known norm $\|\cdot\|$ we are given a set of vectors $X = \{x_i\}_{i=1}^n \subset \mathbb{R}^d$ and an accuracy parameter $\varepsilon \in (0, 1)$, and we must produce some data structure $\mathcal{D}$. Then later, given only $\mathcal{D}$ stored in memory with no direct access to $X$, we must respond to queries that specify $q \in \mathbb{R}^d$ by reporting distance estimates $\tilde{d}_1, \ldots, \tilde{d}_n$ satisfying

$$\forall i \in [n], \ (1 - \varepsilon)\|q - x_i\| \le \tilde{d}_i \le (1 + \varepsilon)\|q - x_i\|.$$

The quantities we wish to minimize in a solution to ADE are (1) pre-processing time (the time to compute $\mathcal{D}$ given $X$), (2) memory required to store $\mathcal{D}$ (referred to as "space complexity"), and (3) query time (the time required to answer a single query). The trivial solution is to, of course, simply store $X$ in memory explicitly, in which case pre-processing time is zero, the required memory is $O(nd)$, and the query time is $O(nd)$ (assuming the norm can be computed in linear time, which is the case for the norms we focus on in this work).

Standard solutions to ADE are via *randomized linear sketching*: one picks a random "sketching matrix" $\Pi \in \mathbb{R}^{m \times d}$ for some $m \ll d$ and stores $y_i = \Pi x_i$ in memory for each $i$. Then to answer a query, $q$, we return some estimator applied to $\Pi(q - x_i) = \Pi q - y_i$. Specifically in the case of $\ell_2$, one can use the Johnson-Lindenstrauss lemma [JL84], AMS sketch [AMS99], or CountSketch [CCF04, TZ12]. For $\ell_p$ norms $0 < p < 2$, one can use Indyk's $p$-stable sketch [Ind06] or that of [KNPW11]. Each of these works specifies some distribution over such $\Pi$, together with an estimation procedure. All these solutions have the advantage that $m = \tilde{O}(1/\varepsilon^2)$, so that the space complexity of storing $y_1, \ldots, y_n$ would only be $\tilde{O}(n/\varepsilon^2)$ instead of $O(nd)$. The runtimes for computing $\Pi x$ for a given $x$ range from $O(d/\varepsilon^2)$ to $O(d)$ for $\ell_2$ and from $O(d/\varepsilon^2)$ to $\tilde{O}(d)$ for $\ell_p$ ([KNPW11]). However, estimating $\|q - x_i\|_p$ from $\Pi q - y_i$ takes $\tilde{O}(n/\varepsilon^2)$ time in all cases. Notably, the recent work of Indyk and Wagner [IW18] is not based on linear sketching and attains the optimal space complexity in bits required to solve ADE in Euclidean space, up to an $O(\log(1/\varepsilon))$ factor.

One downside of all the prior work mentioned in the previous paragraph is that they give Monte Carlo randomized guarantees that do *not* support adaptive queries, i.e. the ability to choose a query vector based on responses by the data structure given to previous queries. Specifically, all these data structures provide a guarantee of the form

$$\forall q \in \mathbb{R}^d, \ \mathbb{P}_s(\text{data structure correctly responds to } q) \geq 1 - \delta,$$

where $s$ is some random "seed", i.e. a random string, used to construct the data structure (for linear sketches specifically, $s$ specifies $\Pi$). The main point is that $q$ is not allowed to depend on $s$; $q$ is first fixed, then $s$ is drawn independently. Thus, in a setting in which we want to support a potentially adversarial sequence of queries $q_1, q_2, \ldots$, where $q_j$ may depend on the data structure's responses to $q_1, \ldots, q_{j-1}$, the above methods do not provide any error guarantees since responses to previous queries are correlated with $s$. Thus, if $q_j$ is a function of those responses, it, in turn, is correlated with $s$. In fact, far from being a technical inconvenience, explicit attacks exploiting such correlations were constructed against all approaches based on linear sketching ([HW13]), rendering them open to exploitation in the adversarial scenario. We present our results in the above context:

**Our Main Contribution.** We provide a new data structure for ADE in the adaptive setting, for $\ell_p$ norms ($0 < p \leq 2$) with memory consumption $\tilde{O}((n+d)d/\varepsilon^2)$, slightly more than the $O(nd)$ required to store $X$ in memory explicitly, but with the benefit that our query time is only $\tilde{O}(\varepsilon^{-2}(n+d))$ as opposed to the $O(nd)$ query time of the trivial algorithm. The pre-processing time is $\tilde{O}(nd^2/\varepsilon^2)$. Our solution is randomized and succeeds with probability $1 - 1/poly(n)$ for each query. Unlike the previous work discussed, the error guarantees hold even in the face of adaptive queries.

In the case of Euclidean space ($p = 2$), we are able to provide sharper bounds with fewer logarithmic factors. Our formal theorem statements appear later as Theorems 4.1 and B.1. Consider for example the setting where $\varepsilon$ is a small constant, like 0.1 and $n > d$. Then, the query time of our algorithm is optimal up to logarithmic factors; indeed just reading the input then writing the output of the distance estimates takes time $\Omega(n + d)$. Secondly, a straightforward encoding argument implies that any such approach must have space complexity at least $\Omega(nd)$ bits (see Section C) which means that our space complexity is nearly optimal as well. Finally, pre-processing time for the data structure can be improved by using fast algorithms for rectangular matrix multiplication (See Section 4 for further discussion).

## 1.1 Related Work

As previously discussed, there has been growing interest in understanding risks posed by the deployment of algorithms in potentially adversarial settings ([BCM$^+$17, HMPW16, GSS15, YHZL19, LCLS17, PMG16]). In addition, the problem of preserving statistical validity in exploratory data

analysis has been well explored [DFH$^+$15a, BNS$^+$16, DFH$^+$15b, DFH$^+$15c, DSSU17] where the goal is to maintain coherence with an unknown distribution from which one obtains data samples. There has also been previous work studying linear sketches in adversarial scenarios quite different from those appearing here ([MNS11, GHR$^+$12, GHS$^+$12]).

Specifically on data structures, it is, of course, the case that deterministic data structures provide correctness guarantees for adaptive queries automatically, though we are unaware of any non-trivial deterministic solutions for ADE. For the specific application of approximate nearest neighbor, the works of [Kle97, KOR00] provide non-trivial data structures supporting adaptive queries; a comparison with our results is given in Subsection 1.2. In the context of streaming algorithms (i.e. sublinear memory), the very recent work of Ben-Eliezer et al. [BEJWY20] considers streaming algorithms with both adaptive queries *and* updates. One key difference is they considered the insertion-only model of streaming, which does not allow one to model computing some function of the difference of two vectors (e.g. the norm of $q - x_i$).

## 1.2 More on applications

**Nearest neighbor search:**   Obtaining efficient algorithms for Nearest Neighbor Search (NNS) has been a topic of intense research effort over the last 20 years, motivated by diverse applications spanning computer vision, information retrieval and database search [BM01, SDI08, DIIM04]. While fast algorithms for *exact* NNS have impractical space complexities ([Cla88, Mei93]), a line of work, starting with the foundational results of [IM98, KOR00], have resulted in query times sub-linear in $n$ for the approximate variant. Formally, the Approximate Nearest Neighbor Problem (ANN) is defined as follows:

**Problem 1.2** (Approximate Nearest Neighbor)**.** Given $X = \{x_i\}_{i \in [n]} \subset \mathbb{R}^d$, norm $\|\cdot\|$, and approximation factor $c > 1$, create a data structure $\mathcal{D}$ such that in the future, for any query point $q \in \mathbb{R}^d$, $\mathcal{D}$ will output some $i \in [n]$ satisfying $\|q - x_i\| \leq c \cdot \min_{j \in [n]} \|q - x_j\|$.

The above definition requires the algorithm to return a point from the dataset whose distance to the query point is close to the distance of the exact nearest neighbor. The Locality Sensitive Hashing (LSH) approach of [IM98] gives a Monte Carlo randomized approach with low memory and query time, but it does not support adaptive queries. There has also been recent interest in obtaining Las Vegas versions of such algorithms [Ahl17, Wei19, Pag18, SW17]. Unfortunately, those works also do not support adaptive queries. More specifically, these Las Vegas algorithms always answer (even adaptive) queries correctly, but their query times are random variables that are guaranteed to be small in expectation only when queries are made non-adaptively.

The algorithms of [KOR00, Kle97] *do* support adaptive queries. However, those algorithms though they have small query time, use large space; [KOR00] uses $\Omega(n^{O(1/\varepsilon^2)})$ space for $c = 1 + \varepsilon$, and [Kle97] uses $\Omega(n^d)$ space. The work of [Kle97] also presents another algorithm with memory and query/pre-processing times similar to our ADE data structure though specifically for Euclidean space. While both of these works provide algorithms with runtimes sublinear in $n$ (at the cost of large space complexity), they are specifically for finding the approximate single nearest neighbor ("1-NN") and do not provide distance estimates to *all* points in the same query time (e.g. if one wanted to find the $k$ approximate nearest neighbors for a $k$-NN classifier).

**Nonparametric estimation:**   While NNS is a vital algorithmic primitive for some fundamental methods in nonparametric estimation, it is inadequate for others, where a few near neighbors do not suffice or the number of required neighbors is unknown. For example, consider the case of kernel regression where a prediction for a query point, $q$, is given by $\hat{y} = \sum_{i=1}^{n} w_i y_i$ where $w_i = K(q,x_i) / \sum_{i=1}^{n} K(q,x_i)$ for some kernel function $K$ and $y_i$ is the label for the $i^{th}$ data point. For this and other nonparametric models including SVMs, distance estimates to potentially every point in the dataset may be required [WJ95, HSS08, Alt92, Sim96, AMS97]. Even for simpler tasks like $k$-nearest neighbor classification or database search, it is often unclear what the right value of $k$ should be and is frequently chosen at test time based on the query point. Unfortunately, modifying previous approaches to return $k$ nearest neighbors instead of 1, results in a factor $k$ increase in query time. Due to the ubiquity of such methods in practical applications, developing efficient versions deployable in adversarial settings is an important endeavor for which an adaptive ADE procedure is a useful primitive.

## 1.3 Overview of Techniques

Our main idea is quite simple and generic, and thus, we believe it could be widely applicable to a number of other problem domains. In fact, it is so generic that it is most illuminating to explain our approach from the perspective of an arbitrary data structural problem instead of focusing on ADE specifically. Suppose we have a randomized Monte Carlo data structure $\mathcal{D}$ for some data structural problem that supports answering nonadaptive queries from some family $\mathcal{Q}$ of potential queries (in the case of ADE, $\mathcal{Q} = \mathbb{R}^d$, so that the allowed queries are the set of all $q \in \mathbb{R}^d$). Suppose further that $l$ independent instantiations of $\mathcal{D}$, $\{\mathcal{D}_i\}_{i=1}^l$, satisfy the following:

$$\forall q \in \mathcal{Q} : \sum_{i=1}^l \mathbf{1} \left\{ \mathcal{D}_i \text{ answers } q \text{ correctly} \right\} \geq 0.9l \qquad \text{(Rep)}$$

with high probability. Since the above holds *for all* $q \in \mathcal{Q}$, it is true even for any query in an adaptive sequence of queries. The Chernoff bound implies that to answer a query successfully with probability $1 - \delta$, one can sample $r = \Theta(\log(1/\delta))$ indices $i_1, \ldots, i_r \in [l]$, query each $\mathcal{D}_{i_j}$ for $j \in [r]$, then return the majority vote (or e.g. median if the answer is numerical and correctness guarantees are approximate). Note that the runtime of this procedure is at most $r$ times the runtime of an individual $\mathcal{D}_i$ and this constitutes the main benefit of the approach: during queries not all copies of $\mathcal{D}$ must be queried, but only a random *sample*. Now, defining for any data structure, $\mathcal{D}$:

$$l^*(\mathcal{D}, \delta) \coloneqq \inf\{l > 0 : \boxed{\text{Rep}} \text{ holds for } \mathcal{D} \text{ with probability at least } 1 - \delta\},$$

the argument in the preceding paragraph now yields the following general theorem:

**Theorem 1.3.** *Let $\mathcal{D}$ be a randomized Monte Carlo data structure over a query space $\mathcal{Q}$, and $\delta \in (0, 1/2)$ be given. If $l^*(\mathcal{D}, \delta)$ is finite, then there exists a data structure $\mathcal{D}'$, which correctly answers any $q \in \mathcal{Q}$, even in a sequence of adaptively chosen queries, with probability at least $1 - 2\delta$. Furthermore, the query time of $\mathcal{D}'$ is at most $O(\log 1/\delta \cdot t_q)$ where $t_q$ is the query time of $\mathcal{D}$ and its space complexity and pre-processing time are at most $l^*(\mathcal{D}, \delta)$ times those of $\mathcal{D}$.*

In the context of ADE, the underlying data structures will be randomized linear sketches and the data structural problem we require them to solve is length estimation; that is, given a vector $v$, we require that at least $0.9l$ of the linear sketches accurately represent its length (See Section 4). In our applications, we select a random sample of $r = \Theta(\log n/\delta)$ linear sketches, use them obtain $r$ estimates of $\|q - x_i\|_p$ and aggregate these estimates by computing their median. The argument in the previous paragraph along with a union bound shows that this strategy succeeds in returning accurate estimates of $\|q - x_i\|_p$ with probability at least $1 - \delta$.

The main impediment to deploying Theorem 1.3 in a general domain is obtaining a reasonable bound on $l$ such that $\boxed{\text{Rep}}$ holds with high probability. In the case that $\mathcal{Q}$ is finite, the Chernoff bound implies the upper bound $l^*(\mathcal{D}, \delta) = O(\log(|\mathcal{Q}|/\delta))$. However, since $\mathcal{Q} = \mathbb{R}^d$ in our context, this is nonsensical. Nevertheless, we show that a bound of $l = \tilde{O}(d)$ suffices for the ADE problem for $\ell_p$ norms with $0 < p \leq 2$ and can be tightened to $O(d)$ for the Euclidean case. We believe that reasonable bounds on $l$ establishing $\boxed{\text{Rep}}$ can be obtained for a number of other applications yielding a generic procedure for constructing adaptive data structures from nonadaptive ones in all these scenarios. Indeed, the vast literature on Empirical Process Theory yields bounds of precisely this nature which we exploit as a special case in our setting.

**Organization:** For the remainder for the paper, we review preliminary material and introduce necessary notation in Section 2, formally present our algorithms in Section 3 and analyze their run time and space complexities in Section 4 before concluding our discussion in Section 6.

## 2 Preliminaries and Notation

We use $d$ to represent dimension, and $n$ is the number of data points. For a natural number $k$, $[k]$ denotes $\{1, \ldots, k\}$. For $0 < p \leq 2$, we will use $\|v\|_p = (\sum_{i=1}^d |v_i|^p)^{1/p}$ to denote the $\ell_p$ "norm" of $v$ (For $p < 1$, this is technically not a norm). For a matrix $M \in \mathbb{R}^{l \times m}$, $\|M\|_F$ denotes the Frobenius norm of $M$: $(\sum_{i,j} M_{i,j}^2)^{1/2}$. Henceforth, when the norm is not specified, $\|v\|$ and $\|M\|$ denote the

standard Euclidean ($p = 2$) and spectral norms of $v$ and $M$ respectively. For a vector $v \in \mathbb{R}^d$ and real valued random variable $Y$, we will abuse notation and use Median($v$) and Median($Y$) to denote the median of the entries of $v$ and the distribution of $Y$ respectively. For a probabilistic event, $\mathcal{E}$, we use $\mathbf{1}\{\mathcal{E}\}$ to denote the indicator random variable for $\mathcal{E}$. Finally, we will use $\mathbb{S}_p^d = \{x \in \mathbb{R}^d : \|x\|_p = 1\}$

One of our algorithms makes use of the $p$-stable sketch of Indyk [Ind06]. Recall the following concerning $p$-stable distributions:

**Definition 2.1.** [Zol86, Nol18] For $p \in (0, 2]$, there exists a probability distribution, Stab($p$), called the $p$-stable distribution with $\mathbb{E}[e^{-itZ}] = e^{-|t|^p}$ for $Z \sim \text{Stab}(p)$. Furthermore, for any $n$, vector $v \in \mathbb{R}^n$ and $Z_1, \dots Z_n$ are iid samples from Stab($p$), we have $\sum_{i=1}^n v_i Z_i \sim \|v\|_p Z$ with $Z \sim \text{Stab}(p)$.

Note the Gaussian distribution is 2-stable, and hence, these distributions can be seen as generalizing the stable properties of a gaussian distribution for norms other than Euclidean. These distributions have found applications in steaming algorithms and approximate nearest neighbor search [Ind06, DIIM04] and moreover, it is possible to efficiently obtain samples from them [CMS76]. We will use $\text{Med}_p$ to denote Median($|Z|$) where $Z \sim \text{Stab}(p)$. Finally, we will use $\mathcal{N}(0, \sigma^2)$ to denote the distribution function of a normal random variable with mean 0 and variance $\sigma^2$.

## 3 Algorithms

As previously mentioned, our construction combines the approach outlined in Subsection 1.3 with known linear sketches [JL84, Ind06] and a net argument. Both Theorem Theorems 4.1 and B.1 are proven using this recipe, with the only differences being a swap in the underlying data structure (or linear sketch) being used and the argument used to establish a bound on $l$ satisfying Rep (discussed in Section 1.3). Concretely, in our solutions to ADE we pick $l = \tilde{O}(d)$ linear sketch matrices $\Pi_1, \dots, \Pi_l \in \mathbb{R}^{m \times d}$ for $m = O(1/\varepsilon^2)$. The data structure stores $\Pi_i x_j$ for all $i \in [l], j \in [n]$. Then to answer a query $q \in \mathbb{R}^d$:

1. Select a set of $r = O(\log n)$ indices $j_k \in [l]$ uniformly at random, with replacement

2. For each $i \in [n], k \in [r]$, obtain distance estimates $\tilde{d}_{i,k}$ based on $\Pi_{j_k} x_i$ (stored) and $\Pi_{j_k} q$. These estimates are obtained from the underlying sketching algorithm (for $\ell_2$ it is $\|\Pi_{j_k} q - \Pi_{j_k} x_i\|_2$ [JL84], and for $\ell_p$ for $0 < p < 2$ it is Median($|\Pi_{j_k} q - \Pi_{j_k} x_i|$)/$\text{Med}_p$ [Ind06] where $|\cdot|$ for a vector denotes entry-wise absolute value.

3. Return distance estimates $\tilde{d}_1, \dots, \tilde{d}_n$ with $\tilde{d}_i = \text{Median}(\tilde{d}_{i,1}, \dots, \tilde{d}_{i,r})$

As seen above, the only difference between our algorithms for the Euclidean and $\ell_p$ norm cases are the distributions used for the $\Pi_j$, as well as the method for distance estimation in Step 2. Since the algorithms are quite similar (though the analysis for the Euclidean case is sharpened to remove logarithmic factors), we discuss the case of $\ell_p$ ($0 < p < 2$) in this section and defer our special treatment of the Euclidean case to Appendix B.

Algorithm 1 constructs the linear embeddings for the case $\ell_p$ norms, $0 < p < 2$, using [Ind06]. The algorithm takes as input the dataset $X$, an accuracy parameter $\varepsilon$ and failure probability $\delta$, and it constructs a data structure containing the embedding matrices and the embeddings $\Pi_j x_i$ of the points in the dataset. The linear embeddings are subsequently used in Algorithm 2 to answer queries.

---

**Algorithm 1** Compute Data Structure ($\ell_p$ space for $0 < p < 2$, based on [Ind06])

---

**Input:** $X = \{x_i \in \mathbb{R}^d\}_{i=1}^n$, Accuracy $\varepsilon \in (0, 1)$, Failure Probability $\delta \in (0, 1)$
$l \leftarrow \Theta\left((d + \log 1/\delta)\log d/\varepsilon\right)$
$m \leftarrow \Theta\left(\frac{1}{\varepsilon^2}\right)$
For $j \in [l]$, let $\Pi_j \in \mathbb{R}^{m \times d}$ with entries drawn iid from Stab($p$)
**Output:** $\mathcal{D} = \{\Pi_j, \{\Pi_j x_i\}_{i=1}^n\}_{j \in [l]}$

---

**Algorithm 2** Process Query ($\ell_p$ space for $0 < p < 2$, based on [Ind06])

---

**Input:** Query $q$, Data Structure $\mathcal{D} = \{\Pi_j, \{\Pi_j x_i\}_{i=1}^n\}_{j \in [l]}$, Failure Probability $\delta \in (0, 1)$
$r \leftarrow \Theta(\log(n/\delta))$
Sample $j_1, \ldots j_r$ iid with replacement from $[l]$
For $i \in [n], k \in [r]$, let $\tilde{d}_{i,k} \leftarrow \frac{\text{Median}(|\Pi_{j_k}(q - x_i)|)}{\text{Med}_p}$
For $i \in [n]$, let $\tilde{d}_i \leftarrow \text{Median}(\{\tilde{d}_{i,k}\}_{k=1}^r)$
**Output:** $\{\tilde{d}_i\}_{i=1}^n$

---

# 4 Analysis

In this section we prove our main theorem for $\ell_p$ spaces, $0 < p < 2$. We then prove Theorem B.1 for the Euclidean case in Appendix B.

**Theorem 4.1.** *For any $0 < \delta < 1$ and any $0 < p < 2$, there is a data structure for the ADE problem in $\ell_p$ space that succeeds on any query with probability at least $1 - \delta$, even in a sequence of adaptively chosen queries. Furthermore, the time taken by the data structure to process each query is $\tilde{O}\left(\varepsilon^{-2}(n+d)\log 1/\delta\right)$, the space complexity is $\tilde{O}\left(\varepsilon^{-2}(n+d)(d+\log 1/\delta)\right)$, and the pre-processing time is $\tilde{O}\left(\varepsilon^{-2}nd(d+\log 1/\delta)\right)$.*

The data structures, $\mathcal{D}_j$, that we use in our instantiation of the strategy described in Subsection 1.3 are the random linear sketches, $\Pi_j$, and the data structural problem they implicitly solve is length estimation; that is, given any vector $v \in \mathbb{R}^d$, at least $0.9l$ of the linear sketches, $\Pi_j$, accurately represent its length. To ease exposition, we will now directly reason about the matrices $\Pi_j$ through the rest of the proof. We start with a theorem of [Ind06]:

**Theorem 4.2** ([Ind06]). *Let $0 < p < 2$, $\epsilon \in (0, 1)$ and $m$ as in Algorithm 1. Then, for $\Pi \in \mathbb{R}^{m \times d}$ with entries drawn iid from Stab$(p)$ and any $v \in \mathbb{R}^d$:*

$$\mathbb{P}\left\{\left(1 - \frac{\varepsilon}{2}\right)\|v\|_p \leq \frac{\text{Median}(|\Pi v|)}{\text{Med}_p} \leq \left(1 + \frac{\varepsilon}{2}\right)\|v\|_p\right\} \geq 0.975$$

One can also find an analysis of Theorem 4.2 that only requires $\Pi$ to be pseudorandom and thus require less memory to store; see the proof of Theorem 2.1 in [KNW10]. We now formalize the Rep requirement on the data structures $\{\Pi_j\}_{j=1}^l$ in our particular context:

**Definition 4.3.** Given $\varepsilon > 0$ and $0 < p < 2$, we say that a set of matrices $\{\Pi_j \in \mathbb{R}^{m \times d}\}_{j=1}^l$ is $(\varepsilon, p)$-*representative* if:

$$\forall \|x\|_p = 1 : \sum_{j=1}^l \mathbf{1}\left\{(1 - \varepsilon) \leq \frac{\text{Median}(|\Pi_j x|)}{\text{Med}_p} \leq (1 + \varepsilon)\right\} \geq 0.9l.$$

We now show that $\{\Pi_j\}_{j=1}^l$, output by Algorithm 1, are $(\varepsilon, p)$-representative.

**Lemma 4.4.** *For $p \in (0, 2)$, $\varepsilon, \delta \in (0, 1)$ and $l$ as in Algorithm 1, the collection $\{\Pi_j\}_{j=1}^l$ output by Algorithm 1 are $(\varepsilon, p)$-representative with probability at least $1 - \delta/2$.*

*Proof.* We start with the simple lemma:

**Lemma 4.5.** *Let $0 < p < 2$, $C > 0$ a sufficiently large constant. Suppose $\Pi \in \mathbb{R}^{m \times d}$ is distributed as follows: the $\Pi_{ij}$ are iid with $\Pi_{ij} \sim$ Stab$(p)$, with $m$ as in Algorithm 1. Then*

$$\mathbb{P}(\|\Pi\|_F \leq C(dm)^{(2+p)/(2p)}) \geq 0.975$$

*Proof.* From Corollary A.2 a $p$-stable random variable $Z$ satisfies $\mathbb{P}(|Z| \geq t) = O(t^{-p})$. Thus by the union bound for large enough constant $C$:

$$\mathbb{P}\left\{\exists i, j : |\Pi_{ij}| \geq C(dm)^{1/p}\right\} \leq \sum_{i,j} \mathbb{P}\left\{|\Pi_{ij}| \geq C(dm)^{1/p}\right\} \leq \frac{1}{40}.$$

Therefore, we have that with probability probability at least 0.975, $\|\Pi\|_F \leq C(dm)^{(2+p)/(2p)}$.

$\square$

Now, let $\theta = C(dm)^{(2+p)/(2p)}$ and define $\mathcal{B} \subset \mathbb{R}^d$ to be a $\gamma$-net (Definition A.3) of the unit sphere $\mathbb{S}_p^d$ under $\ell_2$ distance, with $\gamma = \Theta(\varepsilon(dm)^{-(2+p)/(2p)})$. Recall that $\mathcal{B}$ being a $\gamma$-net under $\ell_2$ means for all $x \in \mathbb{S}_p \exists x' \in \mathcal{B}$ s.t. $\|x - x'\|_2 \leq \gamma$ and that we may assume $|\mathcal{B}| \leq (3/\gamma)^d$ (Lemma A.7).

By Theorems 4.2 and A.8 and our setting of $l$, we have that for any $v \in \mathcal{B}$:

$$\mathbb{P}\left\{ \sum_{j=1}^{l} \mathbf{1}\left\{ (1-\varepsilon/2)\mathrm{Med}_p \leq \mathrm{Median}(|\Pi_j v|) \leq (1+\varepsilon/2)\mathrm{Med}_p \right\} \geq 0.95l \right\} \geq 1 - \frac{\delta}{4|\mathcal{B}|}.$$

Therefore, the above condition holds for all $v \in \mathcal{B}$ with probability at least $1 - \delta/4$. Also, for a fixed $j \in [l]$ Lemma 4.5 yields $\mathbb{P}\left\{ \|\Pi_j\|_F \leq \theta \right\} \geq 0.975$. Thus, by a Chernoff bound, with probability at least $1 - \exp(-\Omega(l)) = 1 - \delta/4$, at least $0.95l$ of the $\Pi_j$ have $\|\Pi_j\|_F \leq \theta$. Thus by a union bound:

$$\forall v \in \mathcal{B} : \sum_{j=1}^{l} \mathbf{1}\left\{ (1-\varepsilon/2)\mathrm{Med}_p \leq \mathrm{Median}(|\Pi_j v|) \leq (1+\varepsilon/2)\mathrm{Med}_p \cap \|\Pi_j\|_F \leq \theta \right\} \geq 0.9l.$$

with probability at least $1 - \delta/2$. We condition on this event and now, extend from $\mathcal{B}$ to the whole $\ell_p$ ball. Consider any $\|x\|_p = 1$. From the definition of $\mathcal{B}$, there exists $v \in \mathcal{B}$ such that $\|x - v\|_2 \leq \gamma$. Let $\mathcal{J}$ be defined as:

$$\mathcal{J} = \{j : (1-\varepsilon/2)\mathrm{Med}_p \leq \mathrm{Median}(|\Pi_j v|) \leq (1+\varepsilon/2)\mathrm{Med}_p \text{ and } \|\Pi_j\|_F \leq \theta\}.$$

From the previous discussion, we have $|\mathcal{J}| \geq 0.9l$. For $j \in \mathcal{J}$:

$$\|\Pi_j x - \Pi_j v\|_\infty \leq \|\Pi_j x - \Pi_j v\|_2 = \|\Pi_j(x - v)\|_2 \leq \|\Pi_j\|_F \|x - v\|_2 \leq \frac{\varepsilon}{2}\mathrm{Med}_p$$

from our definition of $\gamma$ and the bound on $\|\Pi_j\|_F$. Therefore, we have:

$$|\mathrm{Median}(|\Pi x|) - \mathrm{Median}(|\Pi v|)| \leq \frac{\varepsilon}{2}\mathrm{Med}_p.$$

From this, we may conclude that for all $j \in \mathcal{J}$:

$$(1-\varepsilon)\mathrm{Med}_p \leq \mathrm{Median}(|\Pi_j x|) \leq (1+\varepsilon)\mathrm{Med}_p.$$

Since $x$ is an arbitrary vector in $\mathbb{S}_p^d$ and $|\mathcal{J}| \geq 0.9l$, the statement of the lemma follows. $\square$

We prove the correctness of Algorithm 2 assuming that $\{\Pi_j\}_{j=1}^l$ are $(\varepsilon, p)$-representative.

**Lemma 4.6.** *Let $0 < \varepsilon$ and $\delta \in (0, 1)$. Then, Algorithm 2 when given as input any query point $q \in \mathbb{R}^d$ , $\mathcal{D} = \{\Pi_j, \{\Pi_j x_i\}_{i=1}^n\}_{j=1}^l$ where $\{\Pi_j\}_{j=1}^l$ are $(\varepsilon, p)$-representative, $\varepsilon$ and $\delta$, outputs distance estimates $\{\tilde{d}_i\}_{i=1}^n$ satisfying:*

$$\forall i \in [n] : (1-\varepsilon)\|q - x_i\|_p \leq \tilde{d}_i \leq (1+\varepsilon)\|q - x_i\|_p$$

*with probability at least $1 - \delta/2$.*

*Proof.* Let $i \in [n]$ and $W_k = \mathbf{1}\{\tilde{d}_{i,k} \in [(1-\varepsilon)\|q - x_i\|_p, (1+\varepsilon)\|q - x_i\|_p]\}$. We have from the fact that $\{\Pi_j\}_{j=1}^l$ are $(\varepsilon, p)$-representative and the scale invariance of Definition 4.3 that $\mathbb{E}[W_k] \geq 0.9$. Furthermore, $W_k$ are independent for distinct $k$. Therefore, we have by Theorem A.8 that with probability at least $1 - \delta/(2n)$, $\sum_{k=1}^r W_k \geq 0.6r$. From the definition of $\tilde{d}_i$, $\tilde{d}_i$ satisfies the desired accuracy requirements when $\sum_{k=1}^r W_k \geq 0.6r$ and hence, with probability at least $1 - \delta/(2n)$. By a union bound over all $i \in [n]$, the conclusion of the lemma follows. $\square$

Finally, we analyze the runtimes of Algorithms 1 and 2 where $\mathsf{MM}(a, b, c)$ is the runtime to multiply an $a \times b$ matrix with a $b \times c$ matrix. Note $\mathsf{MM}(a, b, c) = O(abc)$, but is in fact lower due to the existence of fast rectangular matrix multiplication algorithms [GU18]; since the precise bound depends on a case analysis of the relationship between $a$, $b$, and $c$, we do not simplify the bound beyond simply stating "$\mathsf{MM}(a, b, c)$" since it is orthogonal to our focus.

**Lemma 4.7.** *The query time of Algorithm 2 is $\tilde{O}((n+d)\log(1/\delta)/\varepsilon^2)$, and for Algorithm 1 the space is $\tilde{O}((n + d)d\log(1/\delta)/\varepsilon^2)$ and pre-processing time is $O(\mathsf{MM}(\varepsilon^{-2}(d + \log(1/\delta))\log(d/\varepsilon), d, n))$ (which is naively $\tilde{O}(nd(d + \log(1/\delta))/\varepsilon^2)$).*

*Proof.* The space required to store the matrices $\{\Pi_j\}_{j=1}^l$ is $O(mld)$ and the space required to store the projections $\Pi_j x_i$ for all $i \in [n], j \in [l]$ is $O(nml)$. For our settings of $m, l$, the space complexity of the algorithms follows. The query time follows from the time required to compute $\Pi_{j_k} q$ for $k \in [r]$ with $r = O(\log n/\delta)$, the $n$ median computations in Algorithm 2 and our setting of $m$. For the pre-processing time, it takes $O(mdl) = \tilde{O}(\varepsilon^{-2}d(d + \log(1/\delta)))$ time to generate all the $\Pi_j$. Then we have to multiply $\Pi_j x_i$ for all $j \in [l], i \in [n]$. Naively this would take time $O(nlmd) = \tilde{O}(\varepsilon^{-2}nd(d+\log(1/\delta)))$. This can be improved though using fast matrix multiplication. If we organize the $x_i$ as columns of a $d \times n$ matrix $A$, and stack the $\Pi_j$ row-wise to form a matrix $\Pi \in \mathbb{R}^{ml \times d}$, then we wish to compute $\Pi A$, which we can do in $\mathsf{MM}(ml, d, n)$ time. $\qquad\square$

**Remark 4.8.** In the case $p = 2$ one can instead use the CountSketch instead of Indyk's $p$-stable sketch, which supports multiplying $\Pi x$ in $O(d)$ time instead of $O(d/\varepsilon^2)$ [CCF04, TZ12]. Thus one could improve the ADE query time in Euclidean space to $\tilde{O}(d + n/\varepsilon^2)$, i.e. the $1/\varepsilon^2$ term need not multiply $d$. Since for the CountSketch matrix, one has $\|\Pi\|_F \le \sqrt{d}$ with probability 1, the same argument as above allows one to establish $(\varepsilon, 2)$-representativeness for CountSketch matrices as well. It may also be possible to improve query time similarly for $0 < p < 2$ using [KNPW11], though we do not do so in the present work.

We now assemble our results to prove Theorem 4.1. The proof of Theorem 4.1 follows by using Algorithm 1 to construct our adaptive data structure, $\mathcal{D}$, and Algorithm 2 to answer any query, $q$. The correctness guarantees follow from Lemmas 4.4 and 4.6 and the runtime and space complexity guarantees follow from Lemma 4.7. This concludes the proof of the theorem.

$\qquad\square$

# 5 Experimental Evaluation

In this section, we provide empirical evidence of the efficacy of our scheme. We have implemented both the vanilla Johnson-Lindenstrauss (JL) approach to distance estimation and our own along with an attack designed to compromise the correctness of the JL approach. Recall that in the JL approach, one first selects a matrix $\Pi \in \mathbb{R}^{k \times d}$ with $k = \Theta(\varepsilon^{-2} \log n)$ whose entries have been drawn from a sub-gaussian distribution with variance $1/k$. Given a query point $q$, the distance to $x_i$ is approximated by computing $\|\Pi(q - x_i)\|$. We now describe our evaluation setup starting with the description of the attack.

**Our Attack:**   The attack we describe can be carried out for any database of at least two points; for the sake of simplicity, we describe our attack applied to the database of three points $\{-e_1, 0, e_1\}$ where $e_1$ is the $1^{\text{st}}$ standard basis vector. Now, consider the set $S$ defined as follows:

$$S := \{x : \|\Pi(x + e_1)\| \le \|\Pi(x - e_1)\|\} = \{x : \langle x, \Pi\Pi^\top e_1 \rangle \le 0\}.$$

When $\Pi$ is drawn from say a gaussian distribution as in the JL-approach, the vector $y = \Pi^\top \Pi e_1$, with high probability, has length $\Omega(\sqrt{d/k})$ while $y_1 \approx 1$. Therefore, when $k \ll d$, the overlap of $y$ with $e_1$ is small (that is, $\langle y, e_1 \rangle / \|y\|$ is small). Conditioned on this high probability event, we sample a sequence of iid random vectors $\{z_i\}_{i=1}^r \sim \mathcal{N}(0, I)$ and compute $z \in \mathbb{R}^d$ defined as:

$$z := \sum_{i=1}^r (-1)^{W_i} z_i \quad \text{where} \quad W_i = \mathbf{1}\left\{\|\Pi(z_i - e_1)\| \le \|\Pi(z_i + e_1)\|\right\}. \tag{1}$$

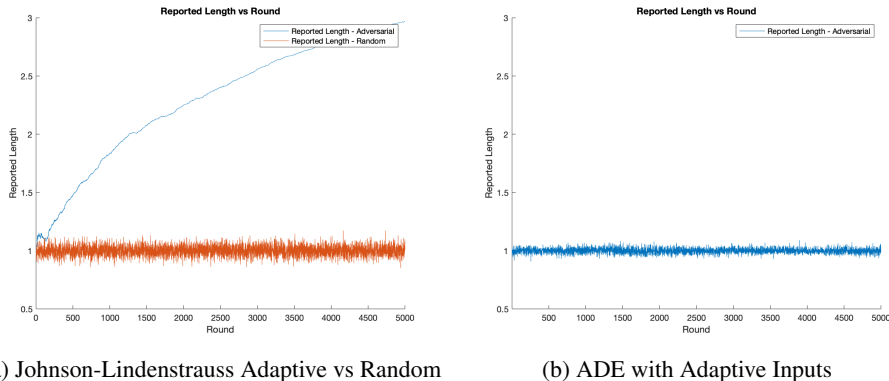

(a) Johnson-Lindenstrauss Adaptive vs Random      (b) ADE with Adaptive Inputs

Figure 1: Subfigure 1a illustrates the impact of adaptivity on the performance of the JL approach to distance estimation and contrasts its performance to what one would obtain if the inputs were random. In contrast, Subfigure 1b shows that the ADE approach described in this paper is unaffected by the attack described here.

Through simple concentration arguments, $z$ can be shown to be a good approximation of $y$ (in terms of angular distance) and noticing that $\|\Pi y\|$ is $\Omega(d/k)$, we get that $\|\Pi z\| \geq \Omega(\sqrt{d/k})\|z\|$ so that $z$ makes a good adversarial query. Note that the above attack can be implemented solely with access to two points from the dataset and the values $\|\Pi(q - e_1)\|$ and $\|\Pi(q + e_1)\|$. Perhaps even more distressingly, the attack consists of a series of *random* inputs and concludes with a *single* adaptive choice. That is, the JL approach to distance estimation can be broken with a *single* round of adaptivity.

In Figure 1, we illustrate the results of our attack on the JL sketch as well as an implementation of our algorithm when $d = 5000$ and $k = 250$ (for computational reasons, we chose a much smaller value of $l = 200$ to implement our data structure). To observe how the performance of the JL approach degrades with the number of rounds, we plotted the reported length of $z$ as in Eq. (1) for $r$ ranging from 1 to 5000. Furthermore, we compare this to the results that one would obtain if the inputs to the sketch were random points in $\mathbb{R}^d$ as opposed to adversarial ones. From Subfigure 1a, the performance of the JL approach drops drastically as as soon as a few hundred random queries are made which is significantly smaller than the ambient dimension. In contrast, Subfigure 1b shows that our ADE data structure is unaffected by the previously described attack corroborating our theoretical analysis.

## 6  Conclusion

In this paper, we studied the problem of adaptive distance estimation where one is required to estimate the distance between a sequence of possibly adversarially chosen query points and the points in a dataset. For the aforementioned problem, we devised algorithms for all $\ell_p$ norms with $0 < p \leq 2$ with nearly optimal query times and whose space complexities are nearly optimal. Prior to our work, the only previous result with comparable guarantees is an algorithm for the Euclidean case which only returns *one* near neighbor [Kle97] and does not estimate all distances. Along the way, we devised a novel framework for building adaptive data structures from non-adaptive ones and leveraged recent results from heavy tailed estimation for one analysis. We now present some open questions:

1. Our construction can be more broadly viewed as a specific instance of *ensemble learning* [Die00]. Starting with the influential work of [Bre96, Bre01], ensemble methods have been a mainstay in practical machine learning techniques. Indeed, the matrices stored in our ensemble have $O(\varepsilon^{-2})$ rows while using a single large matrix would require a model with $O(d)$ rows. Are there other machine learning tasks for which such trade-offs can be quantified?

2. The main drawback our results is the time taken to compute the data structure, $O(nd^2)$ (this could be improved using fast rectangular matrix multiplication, but would still be $\omega(nd)$, i.e. superlinear). One question is thus whether nearly linear time pre-processing is possible.

## 7 Broader Impact

As a theoretical contribution, we do not see our work as having any foreseeable societal implications. While we do provide theoretical insights towards designing more resilient machine learning algorithms, it is difficult to gauge the downstream effects of incorporating such insights into practical algorithms and these effects may heavily rely on context.

## Acknowledgments and Disclosure of Funding

J. Nelson is supported by NSF award CCF-1951384, ONR grant N00014-18-1-2562, ONR DORECG award N00014-17-1-2127, and a Google Faculty Research Award. Y. Cherapanamjeri is supported by a Microsoft Research BAIR Commons Research Grant. The authors would like to thank Sam Hopkins, Sidhanth Mohanty, Nilesh Tripuraneni and Tijana Zrnic for helpful comments in the course of this project and in the preparation of this manuscript.

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
