[Supplementary Material]

# A    Miscellaneous Results and Supporting

## A.1    Properties of Stable Distributions

We will use the following property of stable distributions:

**Lemma A.1.** *[Nol18]  For fixed $0 < p < 2$, the probability density function of a $p$ stable distribution is $\Theta(|x|^{-p-1})$ for large $|x|$.*

By integrating the tail bound from the previous result, we get the following simple corollary.

**Corollary A.2.** *For fixed $0 < p < 2$ and $Z \sim Stab(p)$ and $t$ large:*

$$\mathbb{P}\left\{|Z| \geq t\right\} = \Theta(t^{-p}).$$

## A.2    Probability and High-dimensional Concentration Tools

We recall here standard definitions in empirical process theory from [Ver18].

**Definition A.3** ($\varepsilon$-net [Ver18])**.** Let $(T, d)$ be a metric space, $K \subset T$ and $\varepsilon > 0$. Then, a subset $\mathcal{N} \subset K$ is an $\varepsilon$-net of $K$ if very point in $K$ is within a distance of $\varepsilon$ to some point in $\mathcal{N}$. That is:

$$\forall x \in K, \exists y \in \mathcal{N} : d(x, y) \leq \varepsilon.$$

From this, we obtain the definition of a covering number:

**Definition A.4** (Covering Number [Ver18])**.** Let $(T, d)$ be a metric space, $K \subset T$ and $\varepsilon > 0$. The smallest possible cardinality of an $\varepsilon$-net of $K$ is called the *covering number* of $K$ and is denoted by $\mathcal{N}(K, d, \varepsilon)$.

In the most general set up, we also recall the definition of a covering number.

**Definition A.5** (Packing Number [Ver18])**.** Let $(T, d)$ be a metric space, $K \subset T$ and $\varepsilon > 0$. A subset $\mathcal{P}$ of $T$ is $\varepsilon$-*separated* if for all $x, y \in \mathcal{P}$, we have $d(x, y) > \varepsilon$. The largest possible cardinality of an $\varepsilon$-separated set in $K$ is called the packing number of $K$ and is denoted by $\mathcal{P}(K, d, \varepsilon)$.

We finally recall the following simple fact relating packing and covering numbers.

**Lemma A.6** ([Ver18])**.** *Let $(T, d)$ be a metric space, $K \subset T$ and $\varepsilon > 0$. Then:*

$$\mathcal{P}(K, d, 2\varepsilon) \leq \mathcal{N}(K, d, \varepsilon) \leq \mathcal{P}(K, d, \varepsilon).$$

In all our applications, we will take $d(\cdot, \cdot)$ to be the Euclidean distance and the sets $K$ will always be $\ell_p$ balls for $0 < p \leq 2$. The following lemma follows from a standard volumetric argument.

**Lemma A.7.** *Let $K = \mathbb{S}_p^d$ for $0 < p \leq 2$ and $0 < \varepsilon \leq 1$. Then, we have:*

$$\mathcal{N}(K, \|\cdot\|_2, \varepsilon) \leq \left(\frac{3}{\varepsilon}\right)^d.$$

*Proof.* Note from Lemma A.6 that it is sufficient to prove:

$$\mathcal{P}(K, \|\cdot\|_2, \varepsilon) \leq \left(\frac{3}{\varepsilon}\right)^d.$$

Let $T$ be any $\varepsilon$-separated set in $K$ and let $T_\varepsilon = \{x : \exists y \in T, \|x - y\|_2 \leq \varepsilon/2\}$. Note from the triangle inequality and the fact that $T$ is $\varepsilon$-separated, that for any point $x \in T_\varepsilon$, there exists a unique point $y \in T_\varepsilon$ such that $\|x - y\|_2 \leq \varepsilon/2$. Now, for any point $x \in \mathbb{S}_p^d$, we have:

$$\|x\|_2^2 = \sum_{i=1}^{d} |x_i|^2 \leq \sum_{i=1}^{d} |x_i|^p = 1$$

where the inequality follows from the fact that $|x_i| \leq 1$. Therefore, we have $T \subset \mathbb{B}_2(0, 1, d)$ where $\mathbb{B}_2(x, r, d) = \{y \in \mathbb{R}^d : \|y - x\| \leq r\}$. From this, we obtain from the triangle inequality that $T_\varepsilon \subset \mathbb{B}_2(0, 1 + \varepsilon/2, d)$. From the fact that the sets $\mathbb{B}_2(x, \varepsilon/2, d)$ and $\mathbb{B}_2(y, \varepsilon/2, d)$ are disjoint for distinct $x, y \in T$, we have:

$$\text{Vol}(T_\varepsilon) = |T| \, \text{Vol}(\mathbb{B}_2(0, \varepsilon/2, d)) \leq \text{Vol}(\mathbb{B}_2(0, 1 + \varepsilon/2, d)).$$

By dividing both sides and by using that fact that $\text{Vol}(\mathbb{B}_2(0, l, d)) = l^d \, \text{Vol}(\mathbb{B}_2(0, 1, d))$, we get:

$$|T| \leq \frac{\left(1 + \frac{\varepsilon}{2}\right)^d}{(2/\varepsilon)^d} = \left(1 + \frac{2}{\varepsilon}\right)^d \leq \left(\frac{3}{\varepsilon}\right)^d$$

as $\varepsilon \leq 1$ and this concludes the proof of the lemma. $\qquad\square$

We will also make use of Hoeffding's Inequality:

**Theorem A.8.** *[BLM13] Let $X_1, \ldots, X_n$ be independent random variables such that $X_i \in [a_i, b_i]$ almost surely for $i \in [n]$ and let $S = \sum_{i=1}^{n} X_i - \mathbb{E}[X_i]$. Then, for every $t > 0$:*

$$\mathbb{P}\{S \geq t\} \leq \exp\left(-\frac{2t^2}{\sum_{i=1}^{n}(b_i - a_i)^2}\right).$$

We will also require the bounded differences inequality:

**Theorem A.9.** *[BLM13] Let $\{X_i \in \mathcal{X}\}_{i=1}^{n}$ be $n$ independent random variables and suppose $f : \mathcal{X}^n \to \mathbb{R}$ satisfies the bounded differences condition with constants $\{c_i\}_{i=1}^{n}$; i.e $f$ satisfies:*

$$\forall i \in [n]: \sup_{\substack{x_1,\ldots,x_n \in \mathcal{X} \\ x_i' \in \mathcal{X}}} |f(x_1, \ldots, x_n) - f(x_1, \ldots, x_i', \ldots, x_n)| \leq c_i.$$

*Then, we have for the random variable $Z = f(X_1, \ldots, X_n)$:*

$$\mathbb{P}\{Z - \mathbb{E}[Z] \geq t\} \leq \exp\left(-\frac{t^2}{2v}\right)$$

*where $v = \frac{\sum_{i=1}^{n} c_i^2}{4}$.*

We also present the Ledoux-Talagrand Contraction Inequality:

**Theorem A.10** ([LT11]). *Let $X_1, \ldots, X_n \in \mathcal{X}$ be i.i.d. random vectors, $\mathcal{F}$ be a class of real-valued functions on $\mathcal{X}$ and $\sigma_i, \ldots, \sigma_n$ be independent Rademacher random variables. If $\phi : \mathbb{R} \to \mathbb{R}$ is an $L$-Lipschitz function with $\phi(0) = 0$, then:*

$$\mathbb{E} \sup_{f \in \mathcal{F}} \sum_{i=1}^{n} \sigma_i \phi(f(X_i)) \leq 2L \cdot \mathbb{E} \sup_{f \in \mathcal{F}} \sum_{i=1}^{n} \sigma_i f(X_i).$$

# B  ADE Data Structure for Euclidean Case

---
**Algorithm 3** Compute Data Structure (Euclidean space, based on [JL84])

---
**Input:** Data points $X = \{x_i \in \mathbb{R}^d\}_{i=1}^{n}$, Accuracy $\varepsilon$, Failure Probability $\delta$
$m \leftarrow \Theta\left(\frac{1}{\varepsilon^2}\right), l \leftarrow \Theta\left((d + \log(1/\delta))\right)$
For $j \in [l]$, let $\Pi_j \in \mathbb{R}^{m \times d}$ be such that each entry is drawn iid from $\mathcal{N}(0, 1/m)$
**Output:** $\mathcal{D} = \{\Pi_j, \{\Pi_j x_i\}_{i=1}^{n}\}_{j=1}^{l}$

---

---

**Algorithm 4** Process Query (Euclidean space, based on [JL84])

---

**Input:** Query Point $q$, Data Structure $\mathcal{D} = \{\Pi_j, \{\Pi_j x_i\}_{i=1}^n\}_{j=1}^l$, Failure Probability $\delta$
$r \leftarrow \Theta(\log n + \log 1/\delta)$
Sample $j_1, \ldots j_r$ iid with replacement from $[l]$
For $i \in [n], k \in [r]$, let $y_{i,k} \leftarrow \|\Pi_{j_k}(q - x_i)\|$
For $i \in [n]$, let $\tilde{d}_i \leftarrow \text{Median}(\{y_{i,k}\}_{k=1}^r)$
**Output:** $\{\tilde{d}_i\}_{i=1}^n$

---

In this section we show that logarithmic factors may be improved in an ADE for Euclidean space specifically. Our main theorem of this section is the following.

**Theorem B.1.** *For any $0 < \delta < 1$ there is a data structure for the ADE problem in Euclidean space that succeeds on any query with probability at least $1 - \delta$, even in a sequence of adaptively chosen queries. Furthermore, the time taken by the data structure to process each query is $O\left(\varepsilon^{-2}(n + d) \log n/\delta\right)$, the space complexity is $O\left(\varepsilon^{-2}(n + d)(d + \log 1/\delta)\right)$, and the preprocessing time is $O\left(\varepsilon^{-2}nd(d + \log 1/\delta)\right)$.*

In the remainder of this section, we prove Theorem B.1. We start by introducing the formal guarantee required of the matrices, $\Pi_j$, returned by Algorithm 3:

**Definition B.2.** Given $\varepsilon > 0$, we say a set of matrices $\{\Pi_j \in \mathbb{R}^{m \times d}\}_{j=1}^l$ is $\varepsilon$-representative if:

$$\forall \|v\| = 1 : \sum_{j=1}^l \mathbf{1}\left\{(1 - \varepsilon) \leq \|\Pi_j v\| \leq (1 + \varepsilon)\right\} \geq 0.9l.$$

Intuitively, the above definition states that for any any vector, $v$, most of the projections, $\Pi_j v$, approximately preserve its length. In our proofs, we will often instantiate the above definition by setting $v_i = \frac{q - x_i}{\|q - x_i\|}$, for a query point $q$ and a dataset point $x_i$. As a consequence the above definition, this means that most of the projections $\Pi_j(q - x_i)$ have length approximately $\|q - x_i\|$. By using standard concentration arguments this also holds for the matrices sampled in Algorithm 4 and the correctness of Algorithm 4 follows. The following lemma formalizes this intuition:

**Lemma B.3.** *Let $\varepsilon > 0$ and $0 < \delta < 1$. Then, Algorithm 4, when given as input query point $q \in \mathbb{R}^d$, $\mathcal{D} = \{\Pi_j, \{\Pi_j x_i\}_{i=1}^n\}_{j=1}^l$ for an $\varepsilon$-representative set of matrices $\{\Pi_j\}_{j=1}^l$, $\varepsilon$ and $\delta$ outputs a set of estimates $\{\tilde{d}_i\}_{i=1}^n$ satisfying:*

$$\forall i \in [n] : (1 - \varepsilon)\|q - x_i\| \leq \tilde{d}_i \leq (1 + \varepsilon)\|q - x_i\|$$

*with probability at least $1 - \delta$. Furthermore, Algorithm 4 runs in time $O\left((n + d)m(\log n + \log 1/\delta)\right)$.*

*Proof.* We will first prove that $\tilde{d}_i$ is a good estimate of $\|q - x_i\|$ with high probability and obtain the guarantee for all $i \in [n]$ by a union bound. Now, let $i \in [n]$. From the definition of $\tilde{d}_i$, we see that the conclusion is trivially true for the case where $q = x_i$. Therefore, assume that $q \neq x_i$ and let $v = \frac{q - x_i}{\|q - x_i\|}$. From the fact that $\{\Pi_j\}_{j=1}^l$ is $\varepsilon$-representative, the set $\mathcal{J}$, defined as:

$$\mathcal{J} = \{j : (1 - \varepsilon) \leq \|\Pi_j v\| \leq (1 + \varepsilon)\}$$

has size at least $0.9l$. We now define the random variables $\tilde{y}_{i,k} = \|\Pi_{j_k} v\|$ and $\tilde{z}_i = \text{Median}\{\tilde{y}_{i,k}\}_{k=1}^r$ with $r, \{j_k\}_{k=1}^r$ defined in Algorithm 4. We see from the definition of $\tilde{d}_i$ that $\tilde{d}_i = \|q - x_i\|\tilde{z}_i$. Therefore, it is necessary and sufficient to bound the probability that $\tilde{z}_i \in [1 - \varepsilon, 1 + \varepsilon]$. To do this, let $W_k = \mathbf{1}\{j_k \in \mathcal{J}\}$ and $W = \sum_{k=1}^r W_k$. Furthermore, we have $\mathbb{E}[W] \geq 0.9r$ and since $W_k \in \{0, 1\}$, we have by Hoeffding's Inequality (Theorem A.8):

$$\mathbb{P}\{W \leq 0.6r\} \leq \exp\left(-\frac{2(0.3r)^2}{r}\right) \leq \frac{\delta}{n}$$

from our definition of $r$. Furthermore, for all $k$ such that $j_k \in \mathcal{J}$, we have:

$$1 - \varepsilon \leq \tilde{y}_{i,k} \leq 1 + \varepsilon.$$

Therefore, in the event that $W \geq 0.6r$, we have $(1 - \varepsilon) \leq \tilde{z}_i \leq (1 + \varepsilon)$. Hence, we get:

$$\mathbb{P}\left\{(1 - \varepsilon)\|q - x_i\| \leq \tilde{d}_i \leq (1 + \varepsilon)\|q - x_i\|\right\} \geq 1 - \frac{\delta}{n}.$$

From the union bound, we obtain:

$$\mathbb{P}\left\{\forall i : (1 - \varepsilon)\|q - x_i\| \leq \tilde{d}_i \leq (1 + \varepsilon)\|q - x_i\|\right\} \geq 1 - \delta.$$

This concludes the proof of correctness of the output of Algorithm 4. The runtime guarantees follow from the fact that the runtime is dominated by the cost of computing the projections $\Pi_{j_k} v$ and the cost of computing $\{y_{i,k}\}_{i \in [n], k \in [r]}$ which take time $O(dmr)$ and $O(nmr)$ respectively. $\qquad\square$

Therefore, the runtime of Algorithm 4, is determined by the dimension of the matrices, $\Pi_j$. The subsequent lemma bounds on this quantity as well as the number of matrices, $l$. In our proof of the following lemma, we use recent techniques developed in the context of heavy-tailed estimation [LM19, MZ18] to obtain sharp bounds on both $l$ and $m$ avoiding extraneous log factors.

**Lemma B.4.** *Let $0 < \varepsilon, 0 < \delta < 1$ and $m, l$ be defined as in Algorithm 3. Then, the output $\{\Pi_j\}_{j=1}^l$ of Algorithm 3 satisfies:*

$$\forall \|v\| = 1 : \sum_{j=1}^l \mathbf{1}\left\{(1 - \varepsilon) \leq \|\Pi_j v\| \leq (1 + \varepsilon)\right\} \geq 0.9l$$

*with probability at least $1 - \delta$. Furthermore, Algorithm 3 runs in time $O(\mathsf{MM}(ml, d, n))$.*

*Proof.* We must show that for any $x \in \mathbb{R}^d$, a large fraction of the $\Pi_j$ approximately preserve its length. Concretely, we will analyze the following random variable where $l, m$ are defined in Algorithm 3:

$$Z = \max_{\|v\|=1} \sum_{j=1}^l \mathbf{1}\left\{\left|\|\Pi_j v\|^2 - 1\right| \geq \varepsilon\right\}.$$

Intuitively, $Z$ searches for a unit vector $v$ whose length is well approximated by the fewest number of sample projection matrices $\Pi_j$. We first notice that $Z$ satisfies a bounded differences condition.

**Lemma B.5.** *Let $k \in [l]$, $\Pi_k' \in \mathbb{R}^{m \times d}$ and $Z'$ be defined as:*

$$Z' = \max_{\|v\|=1} \mathbf{1}\left\{\left|\|\Pi_k' v\|^2 - 1\right| \geq \varepsilon\right\} + \sum_{\substack{1 \leq j \leq l \\ i \neq k}} \mathbf{1}\left\{\left|\|\Pi_j v\|^2 - 1\right| \geq \varepsilon\right\}.$$

*Then, we have:*

$$|Z - Z'| \leq 1.$$

*Proof.* Let $Y_j(v) = \mathbf{1}\left\{\left|\|\Pi_j v\|^2 - 1\right| \geq \varepsilon\right\}$ and $Y_k'(v) = \mathbf{1}\left\{\left|\|\Pi_k' v\|^2 - 1\right| \geq \varepsilon\right\}$. The proof follows from the following manipulation:

$$Z - Z' = \max_{\|v\|=1} \sum_{j=1}^{l} Y_j(v) - \max_{\|v\|=1} Y_k'(v) + \sum_{\substack{1 \le j \le l \\ i \ne k}} Y_j(v)$$

$$\le \max_{\|v\|=1} \sum_{j=1}^{l} Y_j(v) - Y_k'(v) - \sum_{\substack{1 \le j \le l \\ i \ne k}} Y_j(v)$$

$$= \max_{\|v\|=1} Y_k(v) - Y_k'(v) \le 1.$$

Through a similar manipulation, we get $Z' - Z \le 1$ and this concludes the proof of the lemma. $\square$

As a consequence of Theorem A.9, it now suffices for us to bound the expected value of $Z$.

**Lemma B.6.** *We have* $\mathbb{E}[Z] \le 0.05l$.

*Proof.* We bound the expected value of $Z$ as follows, using an approach of [LM19] (see the proof of their Theorem 2):

$$\mathbb{E}[Z] \le \frac{1}{\varepsilon} \cdot \mathbb{E}\left[ \max_{\|v\|=1} \sum_{j=1}^{l} |\|\Pi_j v\|^2 - 1| \right]$$

$$\le \frac{1}{\varepsilon} \cdot \left( \mathbb{E}\left[ \max_{\|v\|=1} \sum_{j=1}^{l} |\|\Pi_j v\|^2 - 1| - \mathbb{E}|\|\Pi_j' v\|^2 - 1| \right] + l \max_v \mathbb{E}\left[ |\|\Pi v\|^2 - 1| \right] \right)$$

where $\{\Pi_j'\}_{j=1}^l, \Pi$ are mutually independent and independent of $\{\Pi_j\}_{j=1}^l$ with the same distribution. We first bound the second term in the above display. We have for all $\|v\| = 1$:

$$\mathbb{E}\left[ |\|\Pi v\|^2 - 1| \right] \le \sqrt{\mathbb{E}\left[ (\|\Pi v\|^2 - 1)^2 \right]} = \sqrt{\mathbb{E}[\sum_{i=1}^{m} (\langle w_i, v \rangle^2 - m^{-1})^2]}$$

$$\le \sqrt{\mathbb{E}\left[ \sum_{i=1}^{m} \langle w_i, v \rangle^4 \right]} = \sqrt{\frac{3}{m}}.$$

where $w_i \sim \mathcal{N}(0, I/m)$ are the rows of the matrix $\Pi$. For the first term, we have:

$$\mathbb{E}_{\Pi_j}\left[\max_{\|v\|=1}\sum_{j=1}^{l}|\|\Pi_j v\|^2-1|-\mathbb{E}_{\Pi_j'}\left[|\|\Pi_j' v\|^2-1|\right]\right]$$

$$\leq \mathbb{E}_{\Pi_j,\Pi_j'}\left[\max_{\|v\|=1}\sum_{j=1}^{l}|\|\Pi_j v\|^2-1|-|\|\Pi_j' v\|^2-1|\right]$$

$$=\mathbb{E}_{\Pi_j,\Pi_j',\sigma_j}\left[\max_{\|v\|=1}\sum_{j=1}^{l}\sigma_j\left(|\|\Pi_j v\|^2-1|-|\|\Pi_j' v\|^2-1|\right)\right] \qquad \sigma_j \overset{iid}{\sim}\{\pm 1\}$$

$$\leq 2\mathbb{E}_{\Pi_j,\sigma_j}\left[\max_{\|v\|=1}\sum_{j=1}^{l}\sigma_j|\|\Pi_j v\|^2-1|\right]$$

$$\leq 4\mathbb{E}_{\Pi_j,\sigma_j}\left[\max_{\|v\|=1}\sum_{j=1}^{l}\sigma_j\left(\|\Pi_j v\|^2-1\right)\right] \qquad \text{Theorem A.10}$$

$$=4\mathbb{E}_{\Pi_j,\sigma_j}\left[\max_{\|v\|=1}\sum_{j=1}^{l}\sigma_j\left((\|\Pi_j v\|^2-1)-\mathbb{E}_{\Pi_j'}\left[\|\Pi_j' v\|^2-1\right]\right)\right]$$

$$\leq 4\mathbb{E}_{\Pi_j,\Pi_j',\sigma_j}\left[\max_{\|v\|=1}\sum_{j=1}^{l}\sigma_j\left((\|\Pi_j v\|^2-1)-(\|\Pi_j' v\|^2-1)\right)\right]$$

$$=4\mathbb{E}_{\Pi_j,\Pi_j'}\left[\max_{\|v\|=1}\sum_{j=1}^{l}\left((\|\Pi_j v\|^2-1)-(\|\Pi_j' v\|^2-1)\right)\right]$$

$$\leq 4\mathbb{E}_{\Pi_j}\left[\max_{\|v\|=1}\sum_{j=1}^{l}\left(\|\Pi_j v\|^2-1\right)\right]+4\mathbb{E}_{\Pi_j'}\left[\max_{\|v\|=1}-\sum_{j=1}^{l}\left(\|\Pi_j' v\|^2-1\right)\right]$$

$$\leq 8l\mathbb{E}_{\Pi_j}\left[\left\|\frac{\sum_{j=1}^{l}\Pi_j^\top\Pi_j}{l}-I\right\|\right]\leq \frac{l\varepsilon}{40}$$

where the final inequality follows from the fact that $\frac{\sum_{j=1}^{l}\Pi_j^\top\Pi_j}{l}$ is the empirical covariance matrix of $ml$ standard gaussian vectors and the final result follows from standard results on the concentration of empirical covariance matrices of sub-gaussian random vectors (See, for example, Theorem 4.6.1 from [Ver18]) From the previous two bounds, we conclude the proof of the lemma. $\qquad\square$

Now we complete the proof of Lemma B.4. From Lemmas B.5 and B.6 and Theorem A.9, we have with probability at least $1-\delta$:

$$\forall\|v\|=1:\sum_{j=1}^{l}\mathbf{1}\left\{|\|\Pi_j v\|^2-1|\leq\varepsilon\right\}\geq 0.9l.$$

Now, condition on the above event. Let $\|v\|=1$ and let $\mathcal{J}=\{j:|\|\Pi_j v\|^2-1|\leq\varepsilon\}$. For $j\in\mathcal{J}$:

$$1-\varepsilon\leq\|\Pi_j v\|^2\leq 1+\varepsilon \implies 1-\varepsilon\leq\|\Pi_j v\|\leq 1+\varepsilon.$$

This concludes the proof of correctness of the output of Algorithm 3. The runtime guarantees follow from our setting of $m,l$ and the fact that the runtime is dominated by the time taken to compute $\Pi_j x_i$ for $j\in[l]$ and $i\in[n]$ which can be done by stacking the projection matrices into a single

large matrix $\Pi = [\Pi_1^\top \Pi_2^\top \dots \Pi_l^\top]^\top$ and performing a matrix-matrix multiplication with the matrix containing the data points along the columns.

$\square$

Lemmas B.3 and B.4 now imply Theorem B.1. An algorithm satisfying the guarantees of Theorem B.1 follows by first constructing a data structure, $\mathcal{D}$, using Algorithm 3 with failure probability set to $\delta/2$ and accuracy requirement set to $\varepsilon$. Each query can now be answered by Algorithm 4 with $\mathcal{D}$ by setting the failure probability to $\delta/2$. The correctness and runtime guarantees of this construction follow from Lemmas B.3 and B.4 and the union bound.

$\square$

## C Lower Bound

Here we show that any Monte Carlo randomized data structure for handling adaptive ADE queries in Euclidean space with $> 1/2$ success probability needs to use $\Omega(nd)$ space. Since this will be a lower bound on the space complexity in bits yet thus far we have been talking about vectors in $\mathbb{R}^d$, we need to make an assumption on the precision being used. Fix $\eta \in (0, 1/2)$ and define $B_\eta := \{x \in \mathbb{R}^d : \|x\|_2 \leq 1, \forall i \in [d], x_i \text{ is an integer multiple of } \eta/\sqrt{d}\}$. That is, $B_\eta$ is the subset of the Euclidean ball in which all vector coordinates are integer multiples of $\eta/\sqrt{d}$ for some $\eta \in (0, 1/2)$. We will show that the lower bound holds even in the special case that all database and query vectors are in $B_\eta$.

**Lemma C.1.** $\forall \eta \in (0, 1/2)$, $|B_\eta| = \exp(\Theta(d \log(1/\eta)))$

*Proof.* A proof of the upper bound appears in [AK17]. For the lower bound, observe that if $x_i = c_i \eta/\sqrt{d}$ for $c_i \in \{0, 1, \dots, \lfloor 1/\eta \rfloor\}$, then $\|x\|_2 \leq 1$ so that $x \in B_\eta$. Thus $|B_\eta| \geq \lfloor 1/\eta \rfloor^d$. $\square$

We now prove the space lower bound using a standard encoding-type argument.

**Theorem C.2.** *Fix $\eta \in (0, 1/2)$. Then any data structure for ADE in Euclidean space which always halts within some finite time bound $T$ when answering a query, with failure probability $\delta < 1/2$ and $\varepsilon \in (0, 1)$, requires $\Omega(nd \log(1/\eta))$ bits of memory. This lower bound holds even if promised that all database and query vectors are elements of $B_\eta$.*

*Proof.* Let $\mathcal{D}$ be such a data structure using $S$ bits of memory. We will show that the mere existence of $\mathcal{D}$ implies the existence of a randomized encoding/decoding scheme where the encoder and decoder share a common public random string, with $\mathsf{Enc} : B_\eta^n \to \{0, 1\}^S$. The decoder will succeed with probability 1. Thus encoding length $s$ needs to be at least the entropy of the input distribution, which will be the uniform distribution over $B_\eta^n$, and thus $S \geq \lceil n \log_2 |B_\eta| \rceil$, which is at least $\Omega(nd \log(1/\eta))$ by Lemma C.1.

We now define the encoding: we map $X = (x_i)_{i=1}^n \in B_\eta^n$ to the memory state of the data structure after pre-processing with database $X$ (this memory state is random since the pre-processing procedure may be randomiezd). The encoding length is thus $S$ bits. We now give an exponential-time decoding algorithm which can recover $X$ precisely given only $\mathsf{Enc}(X)$. To decode, we iterate over all $q \in B_\eta$ to discover which $x_i$ equal $q$ (if any). Note $\|q - x_i\|_2 = 0$ iff $q = x_i$, and thus a multiplicative $1 + \varepsilon$-approximation to all distances would reveal which $x_i$ are equal to $q$. To circumvent the nonzero failure probability of querying the data structure, we simply iterate over all possibilities for the random string used by the data structure (since $\mathcal{D}$ runs in time at most $T$ it always flips at most $T$ coins, and there are at most $2^T$ possibilities to check). Since the failure probability is at most $1/2$, the estimate of $q$ to $x_i$ will be zero more than half the time iff $q = x_i$. $\square$