[Reviews · NeurIPS 2020]

Review 1

Summary and Contributions: The paper studies ADE problem: preprocess a set of points p_1,... p_n in R^d so that we can later quickly estimate distances ||q-p_i|| for a given query point q. Usual approach here is to use random dimension reduction methods (and similar) that improve the runtime to O~(n/epsilon^2) for 1+epsilon approximation (assuming d>>1/epsilon^2). A weakness is that all those data structures assume that the queries are not adaptive: the queries do not try to force the data structure to make errors. This paper addresses this weakness, with a nice slim fix, that is actually quite general and can be used in other places.

Strengths: - fundamental problem: ADE is for example one of the standard ways to implement nearest neighbor search, hence improved guarantees here immediately imply improved nearest neighbor search algorithms. - the main idea is quite generic.

Weaknesses: - one could say that the technical novelty is a bit thin (here it is: instead of reducing dimension in O~(1/eps^2) dimensions, reduce to O~(d/eps^2), but then, at query time, just sample a few to estimate the distance). - from practical perspective, I wish the paper presented some more tangible motivation/experiments to quantify the trade-off one has to make between adaptive queries vs some performance loss.

Correctness: yes. no empirical methodology.

Clarity: yes

Relation to Prior Work: yes

Reproducibility: Yes

Additional Feedback:


Review 2

Summary and Contributions: The paper presents a data structure that can output a (1+eps)-approximation to the Euclidean distance from a query point q in R^d to each of n preprocessed points in time O(n/eps^2) per query q, where for simplicity I assume d<n and omit logarithmic factors. The method works not only for Euclidean distances, but for every l_p norm, 0<p<=2. In comparison, a straightforward computation takes time O(nd). The key feature of this randomized data structure is that it works even for an *adaptive* sequence queries. This is trivial for deterministic data structures, but might not hold for randomized ones, including all currently known ones, which are all variants of the JL dimension reduction from dimension d to 1/eps^2. This robustness to adaptive queries is becoming increasingly important, if not crucial, in modern data analysis, which relies heavily on applying such procedures repeatedly and is prone to correlations. The solution is relatively simple technically. The preprocessing stage prepares l^*=O(d) of the aforementioned random projections, and then a query is answered by picking O(log n) out of these l^* stored projections. The analysis uses a union bound over a discretization of all possible queries. This is akin for sampling from a small probability space (e.g., while maintaining pairwise independence). I think this method was previously used by KOR00 and Kle97, in the context of nearest neighbor search, which reports only one point. The paper is completely theoretical and has no experimental evaluation. I have read the author rebuttal. The empirical evaluation seems interesting, but it's too brief and way too late to rely upon. Kernel regression may be a potential application, but this formula relies on k(.,.) which is like inner product rather than distances, for which the ADE provides distortion guarantees. I agree with the comparison to prior work and that the similarity is at a very high level. Overall, my evaluation has not changed.

Strengths: The robustness to adaptive queries is becoming extremely important. The paper presents a simple method to achieve robustness to adaptive queries. It is elegant and clean and may have future applications (although in some sense this technical idea is known). The paper solves all l_p norms, 0<p<=2, using the same technique of p-stable distributions.

Weaknesses: The paper solves a less important problem of estimating the distances from a query q to all the n points (compared say with the closest point). I am not very convinced by the the paper's explanation that it's useful for applications like kernel regression. It's not clear if any of the l_p norms, 0<p<=2, other than Euclidean is very useful in applications (but the case of Euclidean and maybe p=1 are important enough) The paper contains no experimental evaluation, for example to demonstrate where this robustness makes a difference, or that the overhead is small. The advantage in query time would be significant when d is large (say d=sqrt{n}), but then the space complexity O(nd^2) is significant too.

Correctness: It looks correct, and while I did not read the details carefully, the main idea is explained well and should work.

Clarity: The paper is generally well written. However the motivation for this specific problem (ADE) and its applications is not sufficiently convincing.

Relation to Prior Work: This specific problem was not studied before, but the discussion of prior relevant work is adequate.

Reproducibility: Yes

Additional Feedback: L 223: the expression "data structural problem" is awkward The number of references (4 pages) seems large, are all of them needed?


Review 3

Summary and Contributions: This work gives a data structure for distance estimation that supports adaptive queries. These queries must report distance estimates from the query point to all n points in the data structure. The memory requirement Otilde((n+d)d/eps^2) is worse than the O(nd) required for storing the points explicitly. The time requirement is only Otilde((n+d)/eps^2), which is better than the O(nd) required when explicitly storing all points. There is also a preprocessing requirement of Otilde(nd^2/eps^2). Note that known data structures that do not support adaptive queries have roughly the following time/space requirements: Otilde(n/eps^2) memory, Otilde(n/eps^2) time per query. These results are optimal (up to log factors) in standard regimes. For approximate nearest neighbor, data structures that support adaptive queries are known, but require a lot more space.

Strengths: The problem of distance estimation is relevant and results are theoretically grounded. The question of robustness to adaptive queries is a very reasonable question. It has been extensively studied in other models (e.g. online algorithms, streaming,...) and for other questions within the data structure setting. The result achieved appears new. There is a cost of robustness to adaptive queries, but in to put it in perspective, it seems to be less than that for approx nearest neighbor. The main technical idea may be applicable to other problems.

Weaknesses: None.

Correctness: yes

Clarity: yes.

Relation to Prior Work: yes.

Reproducibility: Yes

Additional Feedback:


Review 4

Summary and Contributions: Algorithm which given a data set, constructs a structure which answers distance-estimation queries (in relation to the data) even if the queries are chosen adaptively. The memory needed is larger than just storing the data (additional $d^2$ memory), but the query run-time is less than simple linear scan ($n+d$ instead of $nd$). - Draw random $stab(p)$ matrices (for $p=2$ stab(p) is just normal, for other $p$ this is a generalization of Gaussian 's stability properties) - Proof: the matrices are *representative* (the provide good length estimations w.h.p) - Draw a random sample of the matrices - Proof: by Chernoff the majority vote of the sampled matrices gives good length estimations w.h.p

Strengths: The idea and analysis are interesting and yield a good results.

Weaknesses: The main caveat is that it is that the results seems to guarantee accuracy for every *individual* query and not for a sequence of queries. To obtain such result will usually require the union bound but if the queries are allowed to be chosen adaptively then using the union bound might fail and can not be done without proper justification. It is not clear to me how does the result hold for this case. Edit: the authors addressed my question and explained why the adaptivity holds and my score has been updated accordingly.

Correctness: Not sure (see Weaknesses)

Clarity: The paper is well written and clear

Relation to Prior Work: Absolutely.

Reproducibility: Yes

Additional Feedback:


Review 5

Summary and Contributions: The authors develop an algorithm that efficiently answers distance estimation queries, in the case when the queries are issued adaptively, i.e. depending on the answers to the previous queries. In particular, building on existing approaches they provide a 1+eps- approximation to L^P norm queries with high probability, with 0<p<=2. Their approach is faster than the naive approach, while requiring slightly more memory. Moreover, it has the potential of being applied to many other problems while turning non-adaptive data structures into adaptive ones (provided that some non-trivial properties hold). The authors show that their results are nearly optimal for Euclidean distance (p=2). Overall the paper is very well written, while the results are interesting and novel, as far as I could check.

Strengths: - interesting problem and relevant to the neurips community - provides a general framework that could be applied to other problems - the paper is very well written

Weaknesses: - a large part of the theoretical results are deferred to the appendix

Correctness: All claims seem to be correct, as far as I could check

Clarity: The paper is very well written.

Relation to Prior Work: The results seem to be novel to the best of my knowledge, while the related work is thoroughly discussed.

Reproducibility: Yes

Additional Feedback: - the bulk of their approach lies in ensuring that equation (Rep) be satisfied. It would be nice if the authors could provide in section 1.3 an overview of their technique for achieving that. - algorithm 1: m= O(1/eps^2) -> m = O \tilde (1/eps^2) - line 187 steaming -> streaming - line 247 can’t we just use chernoff bound? (instead of A.8) The following reference is (perhaps marginally) relevant: Moritz Hardt, Jonathan Ullman: Preventing False Discovery in Interactive Data Analysis Is Hard. FOCS 2014: 454-463

[Author Response · NeurIPS 2020]

We thank the reviewers for their careful readings of Submission # 4484, title "On Adaptive Distance Estimation".

**Reviewer 1:** In response to your suggestion, we have implemented a vanilla Johnson-Lindenstrauss (JL) sketch vs.
our structure and shown the results of adaptive querying on both with the following experimental setup:

4 
We have three database vectors $e_1, -e_1, 0$ in 5000 dimensions. We JL sketch down to 250 dimensions. We always query unit norm vectors $q$. We do a sequence of queries, and the $x$-axis specifies which query number we are at in the sequence, and the $y$-axis is the sketch's reported distance to 0 for that query (so, $q$'s length). We do an adaptive attack, where we pick the next query vector from a distribution based on previous queries' distance estimations to $e_1, -e_1$ (attack description included in revision). The orange curve shows the vanilla JL's length estimate, which deviates more from the true length as we do more adaptive queries. The blue curve is our structure's estimates of length, which is correctly always near 1. For our structure, we took the median of 5 randomly selected sketches of 200, which only increased query time by a factor 5.

**Reviewer 2:** Utility for e.g. kernel regression: In kernel regression the database also has $y_i$'s and for a query $q$ we
must (approximately) output $\sum_i k(x, x_i) y_i$ for some kernel $k(\cdot, \cdot)$. For kernels based on Euclidean distances, e.g.
RBF's, it is thus natural to want to have all distances to then approximately compute this sum. We note that for the
slightly similar problem of kernel density estimation, there are *sublinear* time algorithms (e.g. see papers of Charikar
and Siminelakis), but they are not designed to handle adaptive queries, plus we are unaware of similar solutions for
kernel regression. We also remark that there *are* studies of "coreset" constructions for kernel regression which reduce $n$
to some $n' \ll n$ (e.g. (Zheng, Phillips KDD'17)); this is a (weighted) subset of the data that gives approximately the
same answer to any query. Coresets provide the approximation property for all queries and thus support adaptive queries.
Thus the naive $O(nd)$ time query algorithm becomes $O(n'd)$ for adaptive queries. This is orthogonal to our approach
though, and can in fact be combined: one can build our data structure *on the coreset* to get query time $\tilde{O}(n' + d)$.

**Empirical evaluation:** See response to Reviewer 1.

**Novelty compared to [Kle97,KOR00] :** Our work, as well as these two works, all do use the idea of having some
random process (henceforth we will call a "test") that does something useful with good probability for some fixed
vector, amplifying via repetition to work with high probability for all vectors, then doing some form of sampling of
tests at query time. [Kle97,KOR00] descriptions and analyses are tailored to the specific processes, whereas we strive
for a completely general meta theorem (Theorem 1.3) that converts any non-robust structure into a robust one that can
handle adaptive queries; our ADE result is essentially then a corollary. There are other differences; our "tests" are all
different, and furthermore in [KOR00] one test (per level of binary search) is sampled at query time (a "test" there is the
sequence of dot products over $\mathbb{F}_2$, after reducing to Hamming space on the hypercube, of a point with a collection of
random binary vectors from some distribution); sampling only one sketch per query cannot work in our setting unless
we blow the space up by an undesirable $poly(n)/\delta$ factor.

**Reviewer 3:** Thank you for your review.

**Reviewer 4:** We would like to address this reviewer's question regarding correctness. As stated, our results guarantee
correctness with high probability for *any* individual query even in a sequence of adaptively chosen queries. For example,
the $100^{\text{th}}$ query may be chosen depending on the answers to the first 99, but our correctness guarantees still hold. A
union bound does in fact then guarantee correctness for an entire sequence of $T$ adaptive queries with probability
$1 - \delta$ with per query runtime $O(\epsilon^{-2}(n + d) \log(T/\delta))$ by instantiating the data structure with failure probability
parameter $\delta' := \delta/T$. The reason this union bound is permissible: our analysis conditions on a certain event occurring
during the (random) pre-processing stage of the data structure: namely that the set of (random) sketches generated are
"representative" (Definition 4.3). Once we condition on this event of having "representative" sketches, our data structure
actually allows the answering of each subsequent adaptive query in a sequence correctly with 100% success probability,
just by returning the median output of *every* sketch. Doing so though unfortunately leads to slow query time, which
is why our query procedure instead samples only a few ($O(\log(n/\delta))$) random sketches and output the median result
from them. Conditioned on the sketches being representative, this works with high probability even in adaptive settings.

**Reviewer 5:** We thank you for finding typographical errors and for the question about Theorem A.8; in the revision
we will make sure to consistently say "Chernoff bound" in all applications; the generalization to $a_i, b_i$ bounds on the
r.v.'s is indeed not needed for us. We also thank you for the reference to the work of Hardt and Ullman.

[Meta-Review · NeurIPS 2020]

After the rebuttal, four reviewers are in favor of accepting the paper. R2 suggests that an experimental evaluation would strengthen the paper.